# Identification of a Novel Pathogen of Peanut Root Rot, *Ceratobasidium* sp. AG-A, and the Potential of Selected Bacterial Biocontrol Agents

**DOI:** 10.3390/jof11070472

**Published:** 2025-06-21

**Authors:** Ying Li, Xia Zhang, Xinying Song, Manlin Xu, Kang He, Yucheng Chi, Zhiqing Guo

**Affiliations:** 1Shandong Peanut Research Institute, Qingdao 266100, China; li1989ying0921@163.com (Y.L.); zhangxia2259@126.com (X.Z.); songxinying88@126.com (X.S.); xumanlin@126.com (M.X.); sdauhk@163.com (K.H.); 2National Engineering Research Center for Peanut, Qingdao 266100, China

**Keywords:** peanut root rot, *Ceratobasidium* sp. AG-A, biological control, antagonistic bacteria, disease management

## Abstract

Peanut root rot poses a significant threat to global peanut production. In order to identify the new pathogen of peanut root rot in Shandong province, China, and to screen the effective antagonistic biocontrol strains against the identified pathogen, ten symptomatic plants from a peanut field (10% disease incidence) of Rongcheng were sampled for pathogen isolation. The predominant isolate RC-103 was identified as *Ceratobasidium* sp. AG-A through morphological characterization and phylogenetic analysis of ITS and *RPB2* sequences. Pathogenicity was confirmed via Koch’s postulates. Three potent biocontrol strains, namely *Bacillus subtilis* LY-1, *Bacillus velezensis* ZHX-7, and *Burkholderia cepacia* Bc-HN1, were screened for effective antagonism against isolate RC-103 by dual-culture analysis. Their cell suspensions could significantly inhibit the hyphal growth of isolate RC-103, with the percentage inhibition of 54.70%, 45.86%, and 48.62%, respectively. Notably, the percentage inhibition of 10% concentration of the cell-free culture filtrate of *B. subtilis* LY-1 was as high as 59.01%, and the inhibition rate of volatile organic compounds of *B. cepacia* Bc-HN1 was 48.62%. Antagonistic mechanisms primarily involved the induction of hyphal abnormalities. In addition, the culture filtrate of these biocontrol bacteria significantly promoted the growth of peanut and increased the resistance of peanut plants to isolate RC-103, with the biocontrol efficiency reaching 41.86%. In summary, this study identified a novel pathogen of peanut root rot, *Ceratobasidium* sp. AG-A, which was reported for the first time in China, and screened three highly effective antagonistic biocontrol strains against *Ceratobasidium* sp. AG-A isolate RC-103, providing the scientific basis to study the epidemiology and management of this disease.

## 1. Introduction

Peanut (*Arachis hypogaea* L.), a high-protein and high-oil crop, plays a vital role in China’s agricultural economy by ensuring edible oil security and diversifying food sources [1]. China has made remarkable achievements in the field of peanut cultivation, which not only has a vast planting area but also ranks among the best in the world in terms of output [2]. However, peanut root rot is a significant disease that is considered one of the major causes of peanut economic losses worldwide. It has been reported that a variety of pathogens can cause peanut root rot, for example, *Fusarium acuminatum* [3], *Aspergillus niger* [4], *Macrophomina phaseolina* [5] etc., with *Fusarium* spp. being predominant.

At present, *Rhizoctonia* spp. is classified into three groups according to the number of nuclei contained in a single mycelial cell: uninucleate *Rhizoctonia* (UNR), binucleate *Rhizoctonia* (BNR), and multinucleate *Rhizoctonia* (MNR). Among them, isolates of BNR have been categorized into 18 anastomosis groups (AGs) (AG-A to AG-I, AG-K, AG-L, AG-O to AG-S, AG-V, and AG-W), while the MNR was divided into 13 AGs [6,7,8,9]. Moreover, based on the morphological characteristics of sexual generations induced by a few strains, combined with the latest molecular phylogenetic studies, UNR and BNR belong to the genus *Ceratobasidium Rogers*, which is classified as *Ceratobasidiaceae* family in the *Cantharellales* order, *Agaricomycetes* class, *Agaricomycotina* subphylum, *Basidiomycota* phylum [10]. Thus, *Ceratobasidium* sp. are soil-borne pathogens causing critical diseases in economically important plants, including root rot in herbaceous crops [11,12,13]. It causes symptoms such as pre- and post-emergence damping off of *Tagetes erecta* [14], root rot of watermelon [15], stem rot of wishbone flower (*Torenia fournieri*) [16], crown rot of sugar beet [17], basal stem canker of common bean [18], and brown lesions of *Pinellia ternata* (Thumb.) [19].

*Ceratobasidium* spp. can be effectively managed by using chemical fungicides; however, their excessive use may lead to negative impacts on human health and the environment. This prompts agricultural producers to adopt biocontrol strategies as an alternative for sustainable agricultural development [20]. Biological control plays a vital role in reducing plant diseases and maintaining the natural balance of existing ecosystems [21]. At present, the use of biocontrol microorganisms to control root rot is prevalent. Microbes such as *Pseudomonas* [22,23], *Streptomyces* [24], *Trichoderma* [25], and *Bacillus* spp. [26] have been shown to have a strong antagonistic effect in controlling the root rot caused by *Rhizoctonia*. For example, Farhaoui et al. found that only 11 of the 198 strains of biocontrol bacteria had an inhibition rate of over 50% on hyphal growth of *Rhizoctonia*, mainly through the action of metabolites secreted by antagonistic bacteria on mycelial morphology of pathogens [26]. It was found that antifungal compounds produced by *Pseudomonas,* including pyolyteorin, pyrrolnitrin, oligomycin, 2.4 diacetyl phloroglucinol, phenazine, pyocyanin, etc., had a significant inhibitory impact against plant pathogenic fungi [27,28]. Moreover, antagonistic bacteria like *Burkholderia gladioli* strain NGJ1 exhibit mycophagous ability on several fungi by encoding two paralogues of *rpoN*, i.e., *rpoN1* and *rpoN2* [29]. Zhong et al. found that the antibacterial bioactive compounds in the culture filtrate of *Streptomyces olivoreticuli* were stable proteins/peptides, which impaired membrane integrity through membrane lipid peroxidation [30]. In addition, plant extracts were used to control the fungus, such as 1% tannin extract of *Acacia mangium*, which had an inhibition rate of 64.3% against *Ceratobasidium ramicola* [20].

Accurate pathogen identification is paramount for effective disease management, as misdiagnosis exacerbates crop losses [31]. Despite advances, the virulence mechanisms of *Ceratobasidium* sp. and their interactions with biocontrol agents remain underexplored. To the best of our knowledge, this study provided the first report of root rot on peanut caused by *Ceratobasidium* sp. AG-A in China. Our findings enhanced the understanding of pathogen epidemiology and offered critical insights for developing targeted biocontrol strategies against this emerging threat. 

## 2. Materials and Methods

### 2.1. Plant Material and Test Strains

In June 2024, peanut root rot symptoms were observed in Rongcheng City, Shangdong province, China. Ten diseased peanuts sampled from different geographical locations of the affected fields were collected to isolate pathogens. Peanut variety Huayu36 was provided by the Shandong Peanut Research Institute. *Bacillus subtilis* LY-1, *Bacillus velezensis* ZHX-7, and *Burkholderia cepacia* Bc-HN1 were isolated in our laboratory and stored at the General Microbiology Center of the China Microbiological Culture Preservation Management Committee (CGMCC No. 23930, CGMCC No. 20374, and CGMCC No. 28060). The information on these three biocontrol strains is as shown in Appendix A. The strains (*Ceratobasidium* sp. AG-A, AG-C, and AG-K) were provided by Professor Wu Xuehong from China Agricultural University.

### 2.2. Isolations of Ceratobasidium sp.

For pathogen isolation, the roots of peanut plants were washed with tap water, cut into 2–3 mm sections, surface sterilized with 75% ethanol (Lircon, Dezhou, China) for 30 s and 2% sodium hypochlorite (NaClO) (Fangzheng, Tianjin, China) for 5 min, rinsed three times in sterile water, and dried on sterile absorbent paper. Root sections were placed on potato dextrose agar (PDA) medium (Luqiao, Beijing, China) at 25 °C in darkness. After the colonies were developed on PDA, the fungal strains were purified by the hyphal-tip method.

### 2.3. Identification of Ceratobasidium sp. AG-A Isolate RC-103

Morphological identification: The purified isolate RC-103 was cultured at 25 °C for 2–3 d, colony morphology was observed, the mycelia on the colony edge were picked up by an inoculation needle and placed on the slide (Citotest, Haimen, China), and mycelia morphology was observed by microscope (Olympus IX73, Shanghai, China).

The hyphal anastomosis reactions test: The slide confrontation method was adopted [32]. Briefly, test and reference strains were pre-cultured on PDA under dark conditions at 28 °C for 2 d. Mycelial disks (5 mm diameter) excised from colony margins were positioned 2 cm apart on the slides coated with PDA. Slides were incubated in a humid chamber (28 °C, darkness) until hyphal fronts intersected (2–5 mm overlap zone), typically within 2 d. Anastomosis events were subsequently examined using light microscopy.

Molecular identification: The genomic DNA of isolate RC-103 was extracted by the CTAB method, and the internal transcribed spacer (ITS) region and partial fragments of the second largest subunit of RNA polymerase II (*RPB2*) gene were amplified and sequenced with the primer pairs ITS1/ITS4 [33] and RBP2-980F/RPB2-7cR [34]. The amplified products were sent to Beijing Qingke Biological Co., Ltd. (Beijing, China) for sequencing. The sequences obtained in this study were submitted to GenBank (https://www.ncbi.nlm.nih.gov/nuccore/PV669009, accessed on 1 April 2025) (accession numbers PV459990 and PV669009), and a nucleotide BLAST comparison with published sequences at NCBI was made (https://blast.ncbi.nlm.nih.gov/Blast.cgi?PROGRAM=blastn&BLAST_SPEC=GeoBlast&PAGE_TYPE=BlastSearch, accessed on 1 April 2025). A phylogenetic analysis was constructed using the neighbor-joining algorithm of MEGA7.0 software.

### 2.4. Pathogenicity Tests

To prepare the inoculum, twenty Huayu36 peanut seeds were placed in a 500 mL sterile pot with 300 g of autoclaved soil (nutrient soil:vermiculite 2:1), and isolate RC-103 was grown on PDA for 4 d at 25 °C. Two weeks after seedling emergence, ten peanut plants were wounded at the stem base of each plant with a needle and inoculated with one mycelial plug (8 mm in diameter). Peanut plants were treated with a plug of non-colonized PDA in the same way as the control. The seedlings were placed in a plant growth chamber maintained at 25 °C, relative humidity > 70%, 16 h light per day, and irrigated with sterile water. The disease incidence was observed 7 d later. The method described for the isolation of the fungus was used to re-isolate the pathogen from the roots of inoculated plants. This experiment was biologically repeated three times.

### 2.5. Antagonistic Activity of Biocontrol Strains

The experimental methods were as described by Li et al. [35]. Briefly, a bacterial suspension (BS) with an OD_600_ of 0.2 concentration was prepared. In vitro dual-culture analysis involved inoculating pathogen RC-103 plugs with a diameter of 5 mm onto the center of a PDA plate, and 10 μL aliquots of BS at approximately 22 mm away from the fungal plug on four sides; and the same amount of sterile water was used as the control. Each treatment was repeated three times and incubated at 25 °C for 3 d.

The antagonistic activity of volatile organic compounds (VOCs) was determined by the double-dish interlocking method [36]. An aliquot (100 μL) of the BS (OD_600_ = 0.2) was spread on Luria-Bertani (LB) (Solarbio, Beijing, China) plates, and the 5 mm diameter fungal plug was placed in the center of each PDA plate. A non-inoculated LB plate was attached as the control. Each treatment was repeated three times and incubated at 25 °C for 3 d.

After the BS was shaken for 48 h, the supernatant was collected by centrifugation. The cell-free culture filtrate (CF) was obtained by filtration of the supernatant through a 0.22 μm filter membrane (Biosharp, Beijing, China). The CF was mixed at a 10% concentration with PDA culture medium that had been autoclaved and then cooled to about 50 °C, and the CF medium was poured into plates; control plates contained the same volume of water. The 5 mm diameter fungal plug was placed in the center of each PDA plate. Each treatment was repeated three times and incubated at 25 °C for 4 d.

The cross method was used to measure the colony diameter, and the mycelium growth rate method [37] was used to calculate the antagonistic activity. This experiment was biologically repeated three times. The percentage of radial growth inhibition (%) = (D1 − D2)/D1 × 100, where D1 and D2 indicate the growth diameters of isolate RC-103 in the control and treatment plates, respectively.

### 2.6. Effects of Biocontrol Bacteria on the Growth and Disease Level of Peanut

The experimental methods were as described by Li et al. [35]. Briefly, ten-day-old peanut seedlings were selected for growth promotion and biocontrol studies, with three seedlings per pot, for a total of 48 pots. Twelve pots of peanut seedlings were irrigated with the culture filtrate of each bacterial strain (10 mL/pot), and the control (12 pots) was irrigated with LB medium with an equal volume of sterile solution. After one week, the control and treatment peanut plants were divided into two parts. Six pots of peanut seedlings from each group were inoculated with isolate RC-103 mycelia. After one week of inoculation, the disease index and disease control effect were calculated. An additional six pots of peanut seedlings from each group were used to measure fresh and dry weight. The criteria of disease grade referred to Li et al. [35]. This experiment was biologically repeated three times.$$I_{D}=\frac{\sum \left(G_{i}\times N_{i}\right)}{\left(G_{\max}\times N_{T}\right)}\times 100$$

*I_D_*: disease index, *G_i_*: grade (0, 1, 2, 3, 4, 5), *N_i_*: the number of plants at each level, *G_max_*: the highest level (5), *N_T_*: the total number of investigated plants.$$E_{C}=\frac{I_{D}CK-I_{D}T}{I_{D}CK}\times 100\%$$

*E*_c_: efficacy, *I_D_CK*: the control disease index, *I*_D_*T*: the treatment disease index.

### 2.7. Statistical Analysis

All data were subjected to analysis of variance (ANOVA). Duncan’s multiple range test (*p* < 0.05) was used for comparison. Statistical analysis was performed using the Statistical Product and Service Solutions (SPSS) 24 package program. Canvas11 software was used for preparing the figures in this article.

## 3. Results

### 3.1. Disease Assessment

In June 2024, peanut root rot symptoms were observed in Rongcheng City, Shandong Province, China (37°17′12″ N, 122°38′40″ E), with an estimated disease incidence of 10%. Field observations revealed characteristic symptoms: initial chlorosis and wilting of leaves, progressive browning of root tissues, and brown vascular discoloration (Figure 1). Severely affected plants exhibited complete necrosis, accompanied by sloughing of the outer cortical layers and epidermal tissues.

### 3.2. Isolations and Identification of Ceratobasidium sp. AG-A

From 10 diseased peanut roots sampled across distinct geographical locations in affected fields, 10 morphologically similar, rapidly growing fungal colonies were independently isolated within three-day-old culture. The isolate RC-103 was chosen as a representative isolate, which was used for morphological characterization, molecular analysis, phylogenetic analysis, and pathogenicity tests. On PDA, colonies initially appeared white and then turned light gray with fluffy aerial hyphae and no sclerotium formation after 14 d (Figure 2A,B). The fungal colony can cover 9 cm plates after growing on PDA medium for 5 d. Microscopic examination revealed that the septate hyphae were 4.28 to 6.43 μm in width and branched at right angles with a constriction at the origin of the branch point (Figure 2C). These characteristics were consistent with *Ceratobasidium* sp. [38]. When paired with the reference strains, the isolate RC-103 exhibited perfect anastomosis with the AG-A strain (Figure 2D) but failed to exhibit anastomosis with the other AG strains (Figure 2E).

For molecular identification, ITS sequence analysis (GenBank accession no. PV459990) resulted in a 100% match for one accession of *Ceratobasidium* sp. AG-A (MF070683) by BLAST in the NCBI nucleotide database, whereas the *RPB2* sequences (PV669009) showed 96.74% identity with *Ceratobasidium* sp. AG-A (DQ301695). Since the *RPB2* gene sequence was only found in some AGs, the phylogenetic analysis based on ITS sequences was constructed using the neighbor-joining algorithm of MEGA7.0 with all the related AGs of *Ceratobasidium* sp. strains (Figure 3). Another phylogenetic tree combining ITS and *RPB2* sequences is shown in Appendix A. Both results indicate that isolate RC-103 clustered in the same clade with the AG-A strains, which were separated from the other AG strains. Therefore, based on the above experiments, the isolate RC-103 was identified as *Ceratobasidium* sp. AG-A.

### 3.3. Pathogenicity

To assess the virulence of the *Ceratobasidium* sp. AG-A isolate RC-103, pathogenicity assays were conducted under controlled conditions. In the test, after 7 d inoculation, the first symptoms appeared as chlorosis of the lower leaves, the root surface of the inoculation site was dark brown, showing depression, and the longitudinal profile showed vascular bundle necrosis in dark brown (Figure 4). Progressive disease development included multifocal symptoms: crown rot, basal stem cankers, brown root lesions, growth stunting, wilting, and eventual plant mortality. The results indicated that isolate RC-103 had strong pathogenicity. Re-isolation from symptomatic tissues yielded fungal colonies morphologically identical to isolate RC-103, with confirmatory ITS and *RPB2* sequencing. Koch’s postulates were rigorously fulfilled, verifying isolate RC-103 as the etiological agent. These findings conclusively establish *Ceratobasidium* sp. AG-A as a novel pathogen of peanut root rot.

### 3.4. Antagonistic Activity of Biocontrol Bacteria Against Isolate RC-103

The dual-culture analysis was used to measure the inhibition rate of different biocontrol candidates on isolate RC-103, and the results showed that three tested strains (*Bacillus subtilis* LY-1, *Bacillus velezensis* ZHX-7, and *Burkholderia cepacia* Bc-HN1) had antagonistic activity against isolate RC-103 (Figure 5). Furthermore, some biocontrol strains that have no obvious inhibitory effect on mycelial growth have been abandoned. In this study, the functions of these three biocontrol bacteria are mainly analyzed. The BS of the three biocontrol bacteria could significantly inhibit the hyphal growth of isolate RC-103, with the percentage inhibition of 54.70%, 45.86%, and 48.62%. In terms of the inhibition of VOCs generated by the three tested strain cultures on mycelial growth, *B. subtilis* LY-1 and *B. velezensis* ZHX-7 were relatively low, with the inhibition rate of 19.33% and 18.78%, respectively, but the percentage inhibition of *B. cepacia* Bc-HN1 against isolate RC-103 was as high as 48.62%. In addition, we used the CF of 2 d cultures of three biocontrol bacteria incorporated into PDA at a proportion of 10% to test the fungistatic effect of the metabolites of three biocontrol bacteria on isolate RC-103. The results showed that, compared with the control, the CF of *B. subtilis* LY-1 had the most obvious antagonistic activity, with an inhibition rate of 59.01%, followed by *B. cepacia* Bc-HN1 with a inhibition value of 27.03%, and the inhibitory rate of *B. velezensis* ZHX-7 was only 12.16%, which was relatively low.

When observed with a light microscope, the hyphal morphologies of isolate RC-103 in dual culture with the three biocontrol bacteria showed striking differences compared to the control, which was without bacterial antagonism (Figure 6). The exo-metabolites of *B. subtilis* LY-1, *B. velezensis* ZHX-7, and *B. cepacia* Bc-HN1 caused the abnormal mycelium morphology of isolate RC-103 in different ways. As shown in Figure 6, under *B. subtilis* LY-1 treatment, the mycelium of isolate RC-103 showed spiral curvature and twisted folding, while under *B. velezensis* ZHX-7 confrontation, the tip of the mycelium expanded and became spherical. In addition, under *B. cepacia* Bc-HN1 treatment, the mycelium of isolate RC-103 showed an abnormal state, such as twisting and folding.

### 3.5. Biological Efficacy Against Peanut Root Rot

Treatment with biocontrol bacterial culture filtrates significantly enhanced peanut seedling biomass compared to LB medium controls (Figure 7A). *B. subtilis* LY-1 demonstrated the strongest growth promotion, increasing fresh and dry weights by 32.06% and 38.69%, respectively. *B. cepacia* Bc-HN1 and *B. velezensis* ZHX-7 showed intermediate effects, with fresh weight enhancements of 24.89% and 18.29% and dry weight increases of 35.53% and 25.79%, respectively.

In addition, the disease index of peanut seedlings inoculated with biocontrol bacteria and isolate RC-103 was significantly lower than that of control seedlings inoculated with isolate RC-103 alone (Figure 7B). The disease index of the control groups was 86.21, while the disease index of the treatment with three biocontrol bacteria was only 50.12, 58.14, and 54.13. Their corresponding biological control effects reached 41.86%, 32.56%, and 37.21%, respectively (Figure 7B). These results confirm the dual functionality of the tested strains: enhancing plant growth while suppressing the pathogenicity of *Ceratobasidium* sp. AG-A.

## 4. Discussion

Based on the morphological characteristics and phylogenetic analysis of rDNA ITS and *RPB2*, the isolate RC-103 from peanut root was identified as *Ceratobasidium* sp. AG-A, representing the first documented occurrence of this pathogen causing peanut root rot in China. Building upon this taxonomic clarification, we systematically screened and validated three biocontrol bacteria with a highly effective antagonistic effect. These findings provide a scientific foundation for the epidemic and comprehensive control of the disease.

At present, *Rhizoctonia* spp. can infect different parts of the peanut leaf, stem, pod, etc., and cause peanut leaf rot [39], sheath blight [40], damping-off [41], and peanut shell brown spot [42], respectively. The pathogenic fungi causing the above diseases were divided into AG-1-IA, AG-4, and AG-1-IC, and the number of nuclei in these three fungi was more than 4, up to 13, which belonged to MNR. While the fungus of peanut shell brown spot was BNR, which specific AGs it belongs to remains to be further identified [43]. In addition, the above four fungi can produce sclerotia, while isolate RC-103 does not produce sclerotia. The morphology of isolate RC-103 was significantly different from that of the above-mentioned *Rhizoctonia* spp. Therefore, isolate RC-103 was a novel pathogen, and this new discovery had increased the varieties of pathogens causing peanut diseases.

As a plant pathogen, *Ceratobasidium* sp. usually causes stem and root rot. It has been reported in watermelon [15], garlic [44], strawberry [45], *Hedychium coronarium* Koen [46], *Calibrachoa hybrida* [47], and other species. It has also been reported that *Ceratobasidium* sp. (AG-A and AG-G) causes root rot and stem canker of the common bean [18]. Peanuts and the common bean are classified in the leguminous family, so the results provided evidence for the peanut root rot caused by *Ceratobasidium* sp. Meanwhile, this finding expands the known host range of *Ceratobasidium* sp. By monitoring the spread and prevalence of pathogens and through timely prevention and control, we can guarantee high yield and efficiency of crops.

Microorganisms play an important role as biocontrol agents of plant diseases, and good results have been achieved [48,49]. Several bacteria with antagonistic activity are nowadays marketed in numerous countries. Several plant growth-promoting bacteria (PGPB) species, like *B. subtilis*, *B. velezensis*, etc., produce antifungal compounds that inhibit fungal pathogens and compete with them for space and food [49,50]. And *B. cepacia* is known for producing metabolites active against a broad number of pathogenic fungi [51]. However, there are still few studies on the screening of biocontrol strains and biocontrol mechanisms against *Ceratobasidium* sp. In our investigation, we evaluated three isolated biocontrol bacteria as potential antagonistic agents against the mycelial growth of isolate RC-103 using the double culture in vitro, and the inhibition rates were higher than 45%. In previous studies, we revealed that the inhibitory mechanism of *B. subtilis* LY-1 may be through the production of extracellular enzyme activity, including protease, cellulase, amylase, chitinase, and β-1, 3-glucanase, as well as the inclusion of genes encoding iturin, an antifungal lipopeptide [35]. We also confirmed that *B. velezensis* ZHX-7 can secrete protease, cellulase, and amylase [52]. These extracellular enzymes primarily target cell wall components and proteins found in the cell wall matrix of fungi. This explains the phenomenon that these biocontrol bacteria cause abnormal mycelium morphology of pathogenic fungi.

The highest biological control efficacy value of *B. velezensis* SS2 against *R. solani* AG-2–2 in sugar beet was 56.26% [26]. Based on previous studies, the control effect of *B. subtilis* LY-1 against the soil-borne pathogen *Fusarium* spp. was up to 44.71% [35], and the control effect of *B. velezensis* ZHX-7 against *Sclerotium rolfsii*, another important soil-borne disease of peanut, was up to 51.76% [52]. However, in this study, the control efficacy of three biocontrol bacteria against *Ceratobasidium* sp. was relatively low, which may be related to the pathogenicity and infection rate of the pathogen. Although the biocontrol effects of the biocontrol bacteria we screened were not the best, combining both direct growth promotion and broad-spectrum biocontrol activities, they could serve as an ecological solution for controlling soil-borne pathogens, including *Fusarium* spp., *S. rolfsii*, and *Ceratobasidium* sp.

Next, we need to further study the main antagonistic factors of these three biological control strains, as well as the synergistic effects of these antagonistic factors under mixed culture conditions, in order to analyze the biocontrol effect on soil-borne pathogens. The factors influencing rhizosphere colonization will be further analyzed, and the ability of rhizosphere colonization will be improved. This work establishes a critical foundation for integrated management of emerging *Ceratobasidium* sp. threats, bridging the gap between pathogen discovery and sustainable control implementation.

## 5. Conclusions

This study identified *Ceratobasidium* sp. AG-A as a novel causal agent of peanut root rot in China, confirmed through polyphasic characterization (morphological, hyphal anastomosis, molecular, and pathogenic evidence). Three biocontrol strains—*B. subtilis* LY-1, *B. velezensis* ZHX-7, and *B. cepacia* Bc-HN1—demonstrated dual functionality, suppressing pathogen growth through hyphal deformation and enhancing plant biomass. As the first report of *Ceratobasidium* sp. AG-A-induced peanut root rot in China, this work provides critical insights for disease surveillance and sustainable management through targeted biocontrol strategies.

## Figures and Tables

**Figure 1 jof-11-00472-f001:**
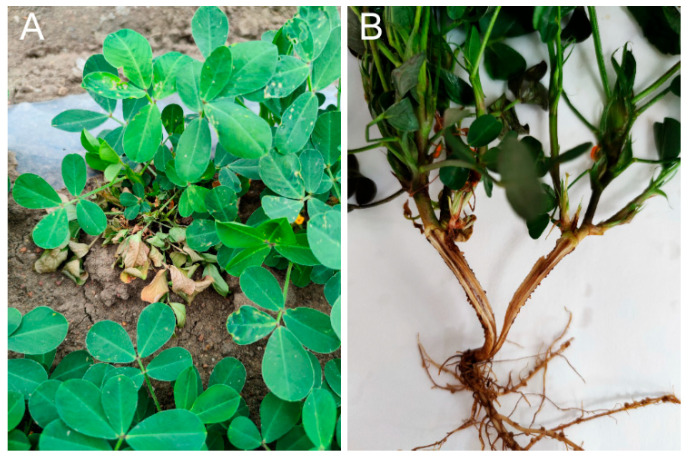
Typical symptoms of peanut root rot. (**A**) Symptoms of diseased plants in the field. (**B**) Internal root infection.

**Figure 2 jof-11-00472-f002:**
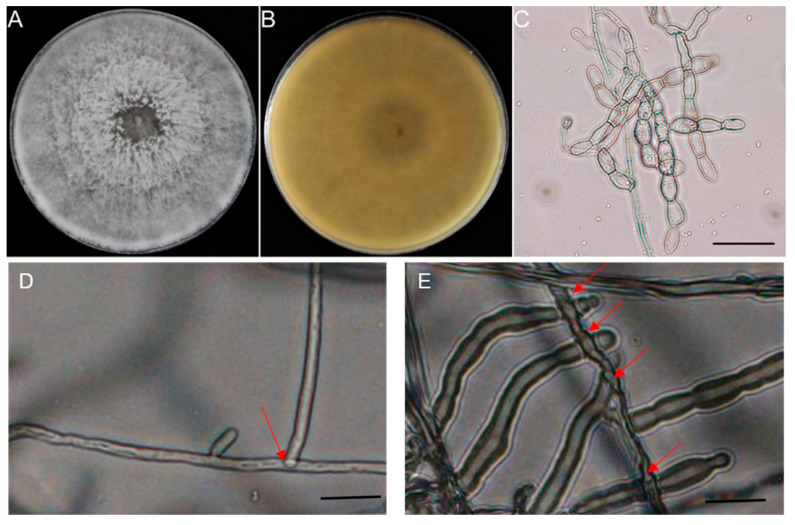
The morphological characterization of isolate RC-103. (**A**,**B**) Pathogen colony morphology. (**C**) Aerial mycelia of isolate RC-103. (**D**) Hyphal anastomosis reaction of isolate RC-103 with *Ceratobasidium* sp. AG-A strain. Arrows indicate the fused hyphae cell. (**E**) Hyphal mismatch reaction of isolate RC-103 with other AG (AG-C, AG-K) strains. The arrows indicate the mycelial cells that cannot fuse. Bar = 20 μm.

**Figure 3 jof-11-00472-f003:**
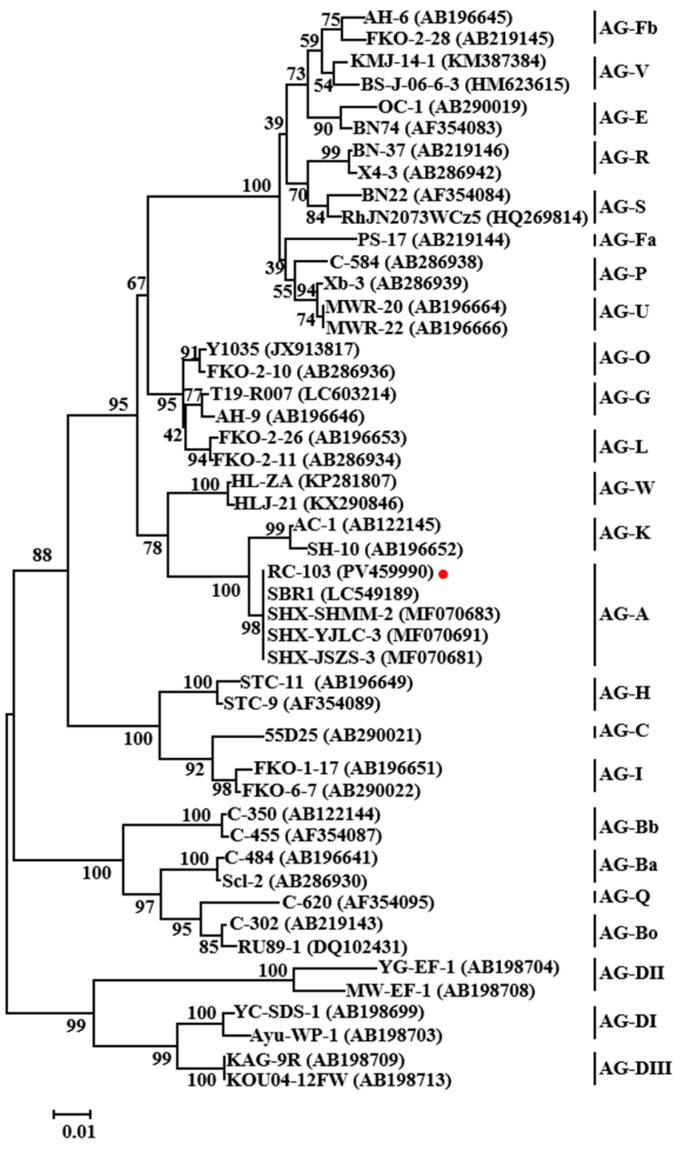
Phylogenetic tree was constructed using neighbor-joining analysis (maximum composite likelihood model in MEGA 7.0) generated from rDNA ITS sequences of isolate RC-103 from this study and other AGs of *Ceratobasidium* sp. from GenBank. The red dot indicates the isolate identified in this study. The numbers on the branch points represented bootstrap values of the tree (1000 replicates). Scale bar represents a genetic distance of 0.01 horizontally. On each branch of the phylogenetic tree is the strain’s name of the *Ceratobasidium* sp., and the GenBank accession number for each strain is provided in the parentheses. The names after the vertical lines represent the anastomosis groups to which the relevant strains belong.

**Figure 4 jof-11-00472-f004:**
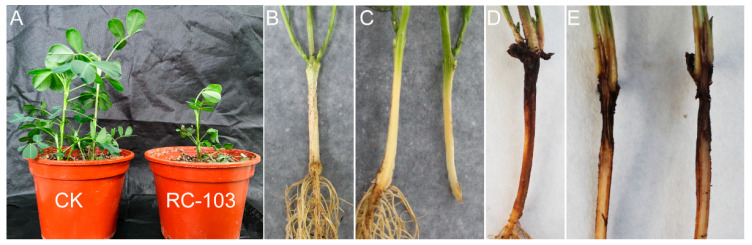
Symptoms caused by *Ceratobasidium* sp. AG-A isolate RC-103 on peanuts. (**A**) Pathogenicity of mycelial plugs inoculated on the root of peanut. (**B**) Healthy root, control, only inoculated PDA medium block, without pathogen inoculation. (**C**) Longitudinal section of healthy root. (**D**) Diseased root, inoculated with the mycelial plugs of isolate RC-103. (**E**) Longitudinal section of diseased root.

**Figure 5 jof-11-00472-f005:**
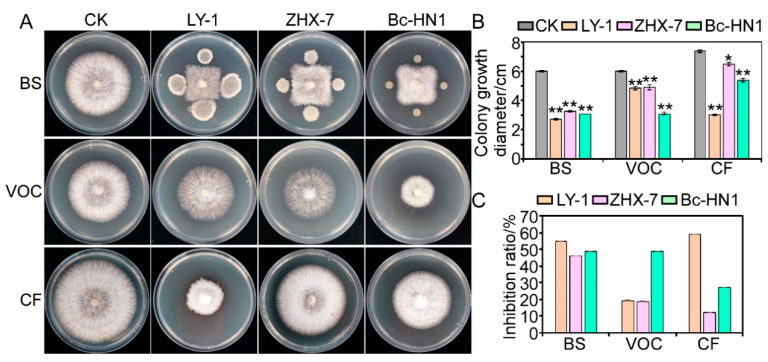
Antagonism of three biocontrol bacteria against isolate RC-103. (**A**) Fungistatic effect of three biocontrol bacteria on isolate RC-103. The fungal colonies of BS and VOC treatments were photographed on 3rd day, and the fungal colonies of CF group were photographed on 4th day. (**B**) Analysis of colony growth of pathogens (error bars = SE). (**C**) Analysis of percentage inhibition by three biocontrol bacteria of growth of isolate RC-103. BS: dual culture of fungal colonies (center), replicate bacterial suspension (periphery); VOC: volatile organic compound; CF: cell-free culture filtrate incorporated in PDA. CK refers to the control, showing growth of isolate RC-103 colonies on the PDA medium without any treatment. ** indicates significant difference from the CK treatment at the *p* < 0.01 level; * indicates significant difference from the CK treatment at the *p* < 0.05 level.

**Figure 6 jof-11-00472-f006:**
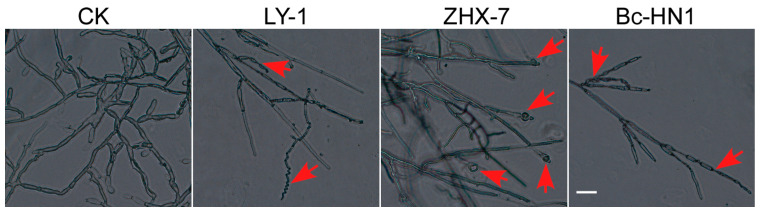
Morphological changes in mycelia of isolate RC-103 upon interaction with three biocontrol bacteria in dual-culture plates. Arrows indicate abnormality and malformation of fungal hypha of isolate RC-103. Bar = 20 μm.

**Figure 7 jof-11-00472-f007:**
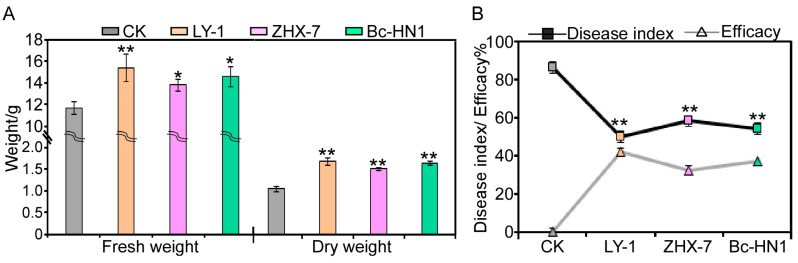
Analysis of growth-promoting and control effects of three biocontrol bacteria on peanut *Ceratobasidium* root rot. (**A**). Fresh mass and dry mass of peanut treated with culture filtrate of three biocontrol bacteria. (**B**). *Ceratobasidium* root rot disease index and disease control effect of each biocontrol bacterium with isolate RC-103 co-inoculation or isolate RC-103 only (CK) on peanut seedlings. ** indicates extremely significant differences at 0.01 level. * indicates significant difference from the CK treatment at the *p* < 0.05 level. Error bars indicate SE.

## Data Availability

The original contributions presented in this study are included in the article/Appendix A. Further inquiries can be directed to the corresponding authors.

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
