# Peer review of "Identification of a Novel Pathogen of Peanut Root Rot, Ceratobasidium sp. AG-A, and the Potential of Selected Bacterial Biocontrol Agents"

_jof, 2025, doi:10.3390/jof11070472_

Round 1
Reviewer 1 Report
It was a pleasure reviewing your work, however, I believe it needs improvement before publication. 1. Do not use keywords that appear in the title as keywords; use other keywords to increase your visibility. 2. What is the objective of your work? Is it unclear? What is your hypothesis? 3. Is the number of replications sufficient for a study of this magnitude? 4. What statistical analysis do you use to compare your treatments? Is it not described? 5. Review all abbreviations used in your work and include their meaning if this is the first time you've used them. Also, indicate their meaning in your results tables. 6. Your conclusion is unclear. Improve it. It was a pleasure reviewing your work, however, I believe it needs improvement before publication. 1. Do not use keywords that appear in the title as keywords; use other keywords to increase your visibility. 2. What is the objective of your work? Is it unclear? What is your hypothesis? 3. Is the number of replications sufficient for a study of this magnitude? 4. What statistical analysis do you use to compare your treatments? Is it not described? 5. Review all abbreviations used in your work and include their meaning if this is the first time you've used them. Also, indicate their meaning in your results tables. 6. Your conclusion is unclear. Improve it.Author Response
Reviewer1:
It was a pleasure reviewing your work, however, I believe it needs improvement before publication.
Comments 1: Do not use keywords that appear in the title as keywords; use other keywords to increase your visibility.
Response 1: Thank you very much for your suggestion. The keywords have been modified to “Peanut root rot; Ceratobasidium sp. AG-A; Biological control; Antagonistic bacteria; Disease management”. These keywords are not entirely derived from the title.
Comments 2: What is the objective of your work? Is it unclear? What is your hypothesis?
Response 2: We are very sorry that we failed to make you understand clearly. Thank you for raising this important question. We acknowledge the necessity to clarify the objective and hypothesis of our research. The following is a concise summary:
- Objectives: “In order to identify the new pathogen of peanut root rot in Shandong province, China, and to screen the effective antagonistic biocontrol strains against the identified pathogen.” This objective is mentioned in lines 12-14 of the abstract part in the text.
- Hypothesis: We hypothesized that the disease outbreak in Shandong was caused by a novel or understudied fungal pathogen, distinct from previously reported peanut root rot pathogens in China. And specific soil-derived bacteria could antagonize the pathogen through direct inhibition (e.g., hyphal disruption) and/or indirect mechanisms (e.g., promote plant growth and enhance the plant's resistance), offering a sustainable management strategy.
Comments 3: Is the number of replications sufficient for a study of this magnitude?
Response 3: The replication strategy in this study was carefully designed to balance scientific rigor with practical feasibility. While our experimental approach included three biological replicates for key assays (pathogenicity tests, antagonistic activity measurements, and biocontrol efficacy evaluations), we acknowledge that expanding replication could enhance statistical power for certain analyses. Specifically:
- Pathogen isolation: Ten diseased plants were sampled from different geographical locations of the affected fields. One isolate was independently obtained from each of these 10 diseased peanuts. The colony morphology of these 10 independent isolates was consistent.
- Pathogenicity tests: Ten peanut plants were inoculated with isolate RC-103 each time, and the tests were repeated three times.
- Biocontrol efficacy evaluations: For each different treatment, 6 pots of peanut seedlings were used, and there were 3 peanut plants in each pot, that is, 18 peanut plants were tested in each treatment.
For studies of this scope investigating novel plant-pathogen interactions, our replication meets minimum requirements in phytopathology research (typically 3-5 biological replicates)[1].
Comments 4: What statistical analysis do you use to compare your treatments? Is it not described?
Response 4: Thank you very much for your reminder. While the original manuscript did not explicitly detail the statistical methods, all the data were analyzed using analysis of variance (ANOVA) and Duncan's multiple comparisons. Statistical analysis was performed using Statistical Product and Service Solutions (SPSS) 24 package program. Canvas11 software was used for making figures in this article. This statistical analysis method has been supplemented in the Materials and Methods section (2.8 Statistical Analysis) in line 195-198.
Comments 5: Review all abbreviations used in your work and include their meaning if this is the first time you've used them. Also, indicate their meaning in your results tables.
Response 5: Thank you. We have carefully inspected and verified.
Comments 6: Your conclusion is unclear. Improve it.
Response 6: Thank you for your suggestion. We have resummarized the conclusion in line 434-442.
Reference:
- McRoberts, N., Hughes, G., and Savary, S., Integrating botanical pesticides in pest management strategies: A challenge for applied research. Phytopathology, 2003. 93: p. S58-S69.
Reviewer 2 Report
This manuscript contains interesting data on Ceratobasidium infecting peanuts. It is certainly of interest of the scientific community working in this area and is
also important as an advance in the peanut industry.
I have a few concerns that I would like the authors to address.
General comments:
1) Is there a way to improve the identification of the pathogen? can the authors include other genes in their phylogenetic analyses?
I am not familiar with Ceratobasidium taxonomy, but ITS is generally not sufficient to identify fungi at the species level - please, provide more information on that. It is not clear why the type strain of different Ceratobasidium species were not included in the analysis. This is crucial to identify the pathogen.
2) Additionally, there is a confusion between Ceratobasidium and Rhizoctonia in the introduction and discussion that needs to be resolved. I found it difficult
to understand. Why the strain obtained by the authors was named AG-A? did the authors perform any anastomosis pairing?
3) Why were these biocontrol agents selected? There is no justification How were the biocontrol agents identified? Is there any sequence data linked to them?
4) How were the carbon sources selected? do they have any significance?
Specific comments:
L11-15 - second and third sentences do not make sense separate.
L20, 25 and elsewhere - Burkholderia cepacia instead of cepacian - please, correct in the whole manuscript.
L75 - please, state the name of the genus - S. olivoreticuli?
L143, 149, … - 600 needs to be subscript
L167 - please, never begin a sentence with a numeral!!!!
L272, 278 - attention to the italics
L313 - what do the authors mean with theoretical basis?
L337-353 - It is not clear what the carbon and nitrogen sources mean for this study and how they help in pathogen characterisation?
Please, comment on the possibility of patenting a potential human pathogen, Burkholderia cepacia.
Round 2
Reviewer 1 Report
The work has been significantly improved, however some modifications are still missing.
Table 1 needs to add the standard error of the mean for its variables.
The work has been significantly improved, however some modifications are still missing.
Table 1 needs to add the standard error of the mean for its variables.
Author Response
The work has been significantly improved, however some modifications are still missing.
Comments 1: Table 1 needs to add the standard error of the mean for its variables.
Response 1: Thank you very much for your comments and support. We have revised and basing on the comments (“I suggest the presentation of Table 1 as a composite figure.”) of the reviewer 2, we replaced Table1 with the corresponding Figure 7.
Reviewer 2 Report
The authors used three bacterial strains that were previously selected in other studies, if I understood correctly. Please, consider the following title for this manuscript:
Identification of a novel pathogen of peanut root rot, Ceratobasidium sp. anastomosis group AG-A, and the potential of selected bacterial biocontrol agents
The introduction still does not provide a clear view of what is Ceratobasidium. Please, add some of the information the authors gave in response to
reviewers on the number of nuclei in different lineages of Rhizoctonia.
In the results section the authors need to present the results on the anastomosis groups, including the results in the text or as a supplementary material if they prefer.
The authors significantly improved the molecular identification, but still did not provide the type strains of the different Ceratobasidium species described in formal taxonomy. This information is essencial.
The biological characterisation of the strain is less interesting than the anastomosis experiments because anastomosis was done in comparison with other strains, whereas the physiological tests were done with strain RC-103 alone. It does not bring any interesting results.
The plate experiments with the biocontrol agents are less interesting than the results with peanut plants, but the authors gave more importance to the plate experiments. I suggest the presentation of Table 1 as a composite figure.
There is still a confusion with the name of the strain and the anastomosis group - please, use Ceratobasidium strain RC-130 anastomosis group AG-A. The anastomosis group does not have to acompany the name of the strain each time.
Line 16 - pathogen instead of pathogens
Line 17 - It is still not clear that AG-A is the anastomosis group. Please, explain that in the text.
Line 33 -…. providing the scientific basis to study the epidemiology ….
Line 48 - why do the authors only add the scientific authorities in some names? Please, add to all or remove from all to be consistent and organised.
Lines 51-52 - This observation causes confusion. Please, note that since 2012 fungi have only one name. Therefore, this observation does not make any sense.
Line 54 - causing instead of caused
Line 60 - spp. instead of sp.
Line 72 - pyrrolnitrin
Line 98 - information instead of informations
Line 103 - PDA medium, not media
Lines 111-117 - please add more information on the anastomosis experiments: which strains were used? where did they come from? This may be provided as
a supplementary table if the authors wish, but I think it is worth showing as a result of the study.
Line 118 - delete biological
Line 130 - what do you mean with seedling sprout? do you mean seedling emergence?
Line 138 - please, add why were these experiments performed. In addition, how did the authors prepare a carbon-free PDA?
Line 155 - ….methods were as described by …. (see other parts of the manuscript, such as line 179).
Line 165 - this is only the control and not control group - please look at other parts of the manuscript as it was used in the whole experiment.
Line 182 - each bacterial strain instead of bacterial kind
Lines190-193 - these formulae could be replaced by disease index (or efficacy) in relation to the control.
Line 198 - ..for preparing the figures.
Line 210 - sp. cannot be in italics - please, verify the whole text.
Lines 236-238 - choose one term to adopt - strain or isolate - they mean the same, but only one should be used throughout the whole manuscript.
Lines 240-242 - some information is missing from this legend: what are the numbers on the branches? what is the scale? which method was used to reconstruct this tree?
which model of evolution was used? where are the type strains? where are the names of the species?
Lines 337-338 - please, improve this legend. Remember that readers should not go back to the methods section to understand what was done here.
Line 347 - please improve.
Line 353 - binucleate is better
Line 355 - this sentence is strange - what do the authors mean with enriched?
Line 356 - sclerotia
Lines 349-360 - this part still needs improvements. It is confusing, specially when you talk about the number of nuclei.
Lines 370-396 - there is too much speculation here. As I mentioned earlier, this physiological characterisation without comparison with other strains does not add any interesting information. Please,
reduce the speculation and the excess of information from the literature that has no direct application to your results.
Line 399 - bacteria
Line 405 - biocontrol twice in the same sentence - please, modify.
